# Lignin Redistribution for Enhancing Barrier Properties of Cellulose-Based Materials

**DOI:** 10.3390/polym11121929

**Published:** 2019-11-22

**Authors:** Wangxia Wang, Tianyu Guo, Kaiyong Sun, Yongcan Jin, Feng Gu, Huining Xiao

**Affiliations:** 1School of Chemistry and Chemical Engineering, Yancheng Institute of Technology, Yancheng 224001, China; wang.w.xia@163.com (W.W.); sunsky@ycit.edu.cn (K.S.); 2Jiangsu Provincial Key Lab of Pulp and Paper Science and Technology, Nanjing Forestry University, Nanjing 210037, China; gty@njfu.edu.cn (T.G.); jinyongcan@njfu.edu.cn (Y.J.); 3Jiangsu R & D Center of the Ecological Dyes and Chemicals, Yancheng Polytechnic College, Yancheng 224005, China; 4Department of Chemical Engineering, University of New Brunswick, Fredericton, NB E3B5A3, Canada

**Keywords:** lignin, redistribution, cellulose, multiple barrier, biodegradability

## Abstract

Renewable cellulose-based materials have gained increasing interest in food packaging because of its favorable biodegradability and biocompatibility, whereas the barrier properties of hydrophilic and porous fibers are inadequate for most applications. Exploration of lignin redistribution for enhancing barrier properties of paper packaging material was carried out in this work. The redistribution of nanolized alkali lignin on paper surface showed excellent water, grease, and water vapor barrier. It provided persisted water (contact angle decrease rate at 0.05°/s) and grease (stained area undetectable at 72 h) resistance under long-term moisture or oil direct contact conditions, which also inhibited the bacterial growth to certain degree. Tough water vapor transmission rate can be lowered 82% from 528 to 97 g/m^2^/d by lignin redistribution. The result suggests that alkali lignin, with multiple barrier properties, has great potential in bio-based application considering the biodegradability, biocompatibility, and recyclability.

## 1. Introduction

Food packaging needs to provide containment and protection for food products during distribution and storage in external and internal unfavorable conditions. Such protection prolongs the shelf life of food products, and meanwhile maintains the optimal quality and safety of the products [1,2,3]. Cellulose-based food packaging material has attracted much attention as a biodegradable and sustainable product compared to petroleum-based plastic [4,5]. However, weak barrier properties to water, grease, and water vapor are critical issues in cellulose-based packaging because of the hydrophilicity and porosity of cellulose fiber network.

Incorporation of nanoparticles within cellulose-based packaging, such as carbon or silica-based ones, either by surface coating or wet-end addition, can enhance the barrier properties of packaging because of tortured channels created for retarding the transfer of water molecules [6]. Alternatively, the nanoparticles could be blended with biodegradable polymers to create nanocomposites for coating. Poly lactic acid (PLA)/montmorillonite composite coating enhanced the water vapor barrier properties of the coated paper significantly. The water vapor permeability of the paper coated with PLA and montmorillonite nanocomposites can be decreased from 300 to 26 g/m^2^/d [7]. However, the introduction of these nanoparticles may affect the recyclability and reusability of cellulose-based packaging materials. Besides, the migration behaviors of nanoparticles in food and their potential impacts on health/safety, as well as the environmental issues need to be addressed [8].

Sustainable and renewable materials, such as starch, chitosan, alginate, and cellulose have been widely used for improving paper-based packaging barrier properties [9,10,11,12]. Cellulose and lignin as the main components of lignocellulose are the most abundant renewable resource on the earth. Micro/nanocellulose exhibits extremely high surface area, lightweight, excellent strength, and good film-forming property, which are widely applied in nanocomposites, papermaking, coating additives, security papers, food packaging, and gas barriers [13,14,15]. Million-fold decline of oxygen permeability coefficient was obtained with only 2% weight loading of ionic liquid-dissolved and -regenerated cellulose [16]. Although polysaccharides have great film-forming property which can help the formation of uniform and dense film on paper surface to resist molecule penetrating through paper, the hydrophilic property of these materials still limits their utilization in dealing with moisture condition.

Alkali lignin degraded from kraft pulping lacks high value utilization because of its complex and condensed structure, while it shows excellent hydrophobicity, antimicrobial, antioxidant, and ultraviolet barrier properties [17,18,19,20,21]. Water vapor transmission rates of the paperboard were significantly decreased with a coat layer of the novel tall oil fatty acid functionalized lignin [22]. Besides, lignin nanoparticles combined with neat or grafted PLA have shown antibacterial activity with reduction in multiplication of the bacterial plant pathogen *Pseudomonas syringae pv. tomato* [23]. These features give lignin great potential to sustainably enhance barrier properties when used as coating, fillers in composites, and as self-standing thin films. In this study, lignin from kraft pulping was reconstructed and redistributed on cellulose-based material as functional barrier for realizing waste lignin valorization and for relieving the environmental problem from petroleum-based materials.

## 2. Materials and Methods

### 2.1. Materials

Fisherbrand™ Qualitative Grade Plain Filter Paper P4 (~98 g/m^2^) and P8 (~68 g/m^2^) was purchased from Fisher Scientific (Ottawa, ON, Canada) and used as base paper for this study. Alkali lignin (AL) was purchased from Sigma Aldrich (St. Louis, MO, USA). Nanofibrillated cellulose (NFC) was obtained from University of Toronto (Toronto, ON, Canada). All the other chemicals were purchased from Sigma Aldrich and used as received without further purification.

### 2.2. Nanolized Alkali Lignin Preparation

Nanolized alkali lignin (NAL) was prepared by an alkali-dissolution and acid-isolation procedure [24]. Briefly, 2 g of alkali lignin was dispersed in 40 mL DI water, and the pH was increased stepwise by adding 1 M NaOH in 1 mL portion under constant stirring. Sufficient equilibrium time between each portion was applied to reach a final target pH value of 11–12. Additional DI water was added to obtain 1 wt % lignin solution and filtrated to remove insoluble particles. Reconstructed NAL was obtained by a gradual decrease of pH with 0.25 M aqueous HCl addition. The final pH value of the colloidal solution was around 2. A dialysis step with 3500 cut-off regenerated cellulose membrane was applied to NAL suspension for removing residual chemicals.

### 2.3. Characterization of NAL

Zeta potential and particle size of NAL were measured using a Zeta potential analyzer (Nanobrook Omni, Brookhaven Instruments Corp., Holtsville, NY, USA). Morphology of NAL was observed with atomic force microscopy (AFM, Nanoscope IIIa, Veeco Instruments Inc., Santa Barbara, CA, USA). One drop of well-dispersed NAL suspension (0.005 wt %) was placed on a clean mica sheet and dried overnight at ambient pressure for morphology analysis. The images were scanned in multimode mode in air using a commercial silicon-tapping probe (NP-S20, Veeco Instruments, Plainview, NY, USA) with a spring constant of 0.32 N/m, a resonance frequency of 273 kHz. Fourier transform infrared spectroscopy (FTIR, Nicolet 6700, Thermo Scientific, Montréal, QC, Canada) spectra were collected between 4000 and 400 cm^−1^ wavenumbers. Thermogravimetric analysis (TGA, Q600, TA Instruments, New Castle, DE, USA) was applied for characterizing thermal stability of NAL. The temperature was controlled from 50 to 800 °C with a heating rate of 10 °C/min under a high purity nitrogen stream.

### 2.4. Lignin Redistribution and Surface Characterization

Base paper (P4/P8 filter paper) was first soaked in distilled water for 24 h before use, and then NFC or NFC/NAL blends (ratio of 1:1) were deposited on paper by vacuum filtration at room temperature (Figure 1). After filtration, the wet paper was stacked between wax paper (top) and paperboard (bottom) to remove water until solid content reach about 80–90%. It was then put into an oven at 60 °C for 24 h along with the two steel plates pressed by 5 kg weight, and then cooled down to room temperature with pressure for another 4 h to avoid deformation. Each deposited paper was prepared five times. Dry weights of filter papers before and after deposition were recorded for calculating deposited weight. Surface morphology of deposited paper was observed by scanning electron microscopy (SEM, JSM-6400, JEOL, Tokyo, Japan). Air permeability of deposited paper was measured according to TAPPI standard method (T 460 om-06) using Gurley air permeability tester 4110 (Teledyne Gurley, Troy, NY, USA), which calculated the time required for a certain volume of air to pass through a test specimen [25].

### 2.5. Water Contact Angle Measurement

The hydrophobic properties of deposited paper were investigated by water contact angle measurements, which were conducted with the sessile drop test method using a versatile optical tensiometer (Theta Attension Tensiometer, Attension/Biolin Scientific, Espoo, Finland). A drop of distilled water (3 μL) was deposited on the surface of the test specimen. Each sample was measured on five different points and the average values were obtained.

### 2.6. Grease Resistance Measurement

Grease barrier properties of deposited papers against vegetable oil were conducted according to TAPPI standard (T507 cm-99) [26]. In order to get more accurate results, a scanner (CanoScan, LiDE 700F, Canon, Tokyo, Japan) was employed to check the grease stained area in this study. Assemblies prepared with bed plate, foil separator, stain absorber, test specimen, and saturated blotter, covered with the pressure block (400 g), from bottom to top, were placed in the oven for 4–72 h at 60 °C. The grease-stained blotter paper was scanned every 4 h if necessary. Finally the percentage of the stained area were calculated and reported. The results were reported as the average of two measurements.

### 2.7. Water Vapor Resistance Measurement

Water vapor transmission rate (WVTR), as the steady water vapor flow in per unit time through unit area of paper under specific conditions of temperatures and humidity was applied for characterizing water vapor barrier property of deposited paper. The WVTR measurements were replicated two times and carried out at 23 °C and 50% of the relative humidity (RH%, achieved by saturated magnesium nitrate salt) according to TAPPI standard (T448om-09) [27]. The round deposited paper samples were clamped in a permeation cell with salt solution. After the permeation cell was placed in an anhydrate chamber, the data was collected after 0.5 h to allow the transmission to reach a steady state. At constant temperature and RH (%), WVTR can be calculated from the change in the weight of the container, at a specified time interval, and the area of exposed coated paper, as described by Equation (1):(1)$$WVTR=\frac{Weight\;Change\;\left(g\right)}{Area\left(m^{2}\right)\times time\left(d\right)}$$

### 2.8. Antibacterial Activity of Deposited Paper

Liquid medium test was carried out for testing the antibacterial activity of deposited paper against Gram-negative bacterial *E. coli* ATCC 11229 [28]. Briefly, 0.1 g of paper scraps and 0.1 mL of bacterial culture (10^10^ CFU/mL) were mixed in nutrient broth and incubated at 37 °C, 100 rpm for 24 h. Then 0.175 mL of diluted culture (10^−4^) was seeded on the agar plate and incubated at 37 °C for 24 h. The number of colonies was counted by Image J software (Image J 1.52, National Institutes of Health, Bethesda, MA, USA), and three repeats were conducted for each sample. The inhibition of cell growth can be quantified by Equation (2):(2)$$Growth\;inhibition\;of\;cell\;\left(\%\right)=\frac{\left(A-B\right)}{A}\times 100\%$$
where *A* and *B* are the number of colonies determined from the blank and treated samples, respectively.

## 3. Results and Discussion

### 3.1. Nanocellulose Fractionation

#### Characterization of NAL

Lignin nanoparticles with uniform size were obtained via an alkali-solution and acid-isolation method. The average diameter was 51.6 ± 2.1 nm, measured by Image J software based on AFM imaging (Figure 2a), which was 55.4 ± 1.4 nm as determined using a Zeta potential analyzer. The zeta potential of NAL was −44.35 ± 0.7 mV. A higher surface charge tends to result in a more stable suspension system, which might benefit for the formation of uniform and dense film during vacuum deposition [29]. FTIR spectra of NAL in spectral range 750–1800 cm^−1^ are shown in Figure 2b. The band near 1510 cm^−1^ was assigned to phenolic ring vibration and the bands at 1450, 1210, and 1188 cm^−1^ was assigned to the vibration of phenolic ring or phenolic hydroxyl groups. Decreasing aryl-ether bonds can result in dropping the bands at 1140, 1050, and 1030 cm^−1^. The increasing intensity of band at 1510 cm^−1^ and decreasing intensity of band at 1140 cm^−1^ compared with Björkman lignin suggest the cleavage of aryl-ether bonds, which brings an increase of phenolic hydroxyl content of alkali lignin after kraft cooking [30]. The increasing phenolic hydroxyl group of NAL benefits the formation of hydrogen bonding with cellulose, which will help expose the hydrophobic side of lignin for water barrier.

Thermogravimetric analysis of NAL is shown in Figure 2c. The initial pyrolysis stage (about 3% weight loss) from 100 to 150 °C was mainly attributed to the moisture release [31]. About 50% of lignin was degraded from 150 to 600 °C, and continue with very slow degradation rate. NAL showed an extremely wide decomposition temperature range. The highest decomposition rate at 344 °C indicated the cleavage of C–C bonds and degradation of aromatic rings. Weight loss from 450 to 800 °C was due to the demethoxylation and re-condensation of volatile products, which contributed to the formation of char [32]. The highly thermal stability of lignin compared with cellulose has great advantage under high operating temperature.

### 3.2. Surface Properties of Deposited Paper

Surface morphologies of non-deposited (control) and deposited papers were characterized by SEM (Figure 3). After NFC + NAL blends deposition, most interfibrous voids, either from fine fiber formed P4 paper or from coarse fiber formed P8 paper, can be filled or covered. The coverage was confirmed by the lowered air permeability from 3.9 ± 0.3 to 1610 ± 8 s/100 mL. Lignin adhering to fiber surface also can be observed clearly (Figure 2f). Figure 1 illustrates the interaction between lignin and cellulose fiber after deposition. Lignin could fill the gap between fibers for reducing porosity. At the meantime, lignin could cover hydrophilic cellulose and expose hydrophobic surface for resisting moisture. Besides, hydrogen bonds formed among paper fiber, NFC and NAL can effectively shield against molecules. Uniform and dense film formed on paper the surface with hydrophobic nanoparticle will provide excellent barrier properties, especially for water and water vapor (Figure 3e).

### 3.3. Water Resistance of Deposited Paper

There were two potential flow paths for water to flow along: channels formed by fiber overlap (the major flow path) and crevices formed by indentations and surface roughness on the fiber walls [33]. The addition of fillers could affect the pore structure and impact the imbibition rate of liquids into paper. Besides, numerous studies have confirmed that combination of micrometer-scale and nanometer-scale roughness, along with a low surface energy material, leads to apparently a highly water contact angle (WCA) [34,35]. The decrease rate of WCA represents a dynamic process of water spreading on paper surfaces. As we can see from Figure 4, NAL can effectively improve the water resistance of deposited paper. It was found that the WCA reduction rate over the range of 30 s to 300 s for P4 (NFC + NAL) was only 0.05 ± 0.01 °/s, which was 0.16 ± 0.01 °/min for 7.6 g/m^2^ NFC deposited one over the same range. Similar results can be found for P8 filter paper as shown in Figure 4b. This result demonstrated that the introduction of NAL gave persisted water resistance under direct moisture contact condition.

### 3.4. Grease Resistance of Deposited Paper

Grease stained areas were detected for characterizing grease resistance of deposited paper (Table 1). Certain amount of nano-sized particles helps reduce paper porosity and obstruct molecular pathway [36]. Nanocellulose deposition (3.6 g/m^2^) can decrease grease stained area from 90.5% to 37.1%. Lignin redistribution on paper surface showed excellent grease barrier. Grease stained area can be further decreased from 37.1% to undetectable (complete grease resistance) with lignin addition.

### 3.5. Water Vapor Resistance of Deposited Paper

It has been established that pores between fiber and fiber are primarily responsible for controlling the water vapor barrier property of paper [37]. Diameter of water molecule is less than 1 nm, which makes the water vapor resistance becoming the top bottleneck for paper-based food packaging. In this study, the results showed that WVTR could be lowered by increasing deposited weight of nanofibers (Figure 5a). The WVTR of control P4 paper was 528 ± 15 g/m^2^/d. When loading 3.6 g/m^2^ NFC, WVTR can be reduced to 419 ± 12 g/m^2^/d, which can be further reduced to 147 ± 10 g/m^2^/d at deposited weight of 18 g/m^2^.

WVTR results of introducing NAL to NFC deposited paper demonstrated that NAL could improve paper water vapor resistance effectively (Figure 5b). WVTR value of NFC + NAL deposited P4 paper (3.6 g/m^2^ NFC and 3.6 g/m^2^ NAL) was 97 ± 12 g/m^2^/d, which were 419 ± 12 g/m^2^/d, 320 ± 11 g/m^2^/d and 147 ± 10 g/m^2^/d for 3.6 g/m^2^, 7.2 g/m^2^ and 18 g/m^2^ NFC deposition without lignin. Lowered WVTR by lignin redistribution might attribute to the fact that the water absorbed on cellulose still can transfer through capillary action, while hydrophobic lignin can complete obstruct moisture flow through. Besides, hydrogen bond formed between nanofibrillated cellulose for resisting water and water vapor could be easily disrupted by water molecules [38]. Although the nanolized hydrophobic lignin enhanced paper barrier properties, the physical hydrogen bond formed between NFC and NAL might still relatively robust because of the branched structure of NAL. Thus, the influence of lignin-coated paper in the presence of crosslinking agent on paper barrier properties will be investigated in our future study.

### 3.6. Antibacterial Activity of Deposited Paper

The application of lignin may allow the synthesis of nanoparticles with biodegradable cores that have higher antimicrobial activity and less environmental impact than metallic silver nanoparticles [39]. Many recent researches reported that lignin showed antibacterial properties in biocomposites [40,41,42]. The antibacterial activity of NAL redistributed filter paper was assessed in this work. According to the growth cell numbers of control paper (Figure 6a) and NAL redistributed filter paper (Figure 6b), NAL did retard the growth of bacterium (i.e., *E. coli*). The inhibition of cell growth by NAL redistribution was 77% ± 6%. The antibacterial activity can be further improved to over 99% by the addition of the guanidine-based starch (ATPS) according to our previous study [43].

## 4. Conclusions

Lignin redistribution for enhancing water, grease, and water vapor barrier properties of hydrophilic/porous cellulose-based packaging material was investigated in this work. Excellent barrier properties of deposited paper were achieved with nanolized hydrophobic lignin deposition by pore filling and hydrogen bond formation. Initial water contact angle was over 100° with decreased rate only at 0.05°/s for NAL redistributed paper. Besides, complete grease resistance (100%) and much lowered water vapor transmission rate (82%) were achieved. Whole lignocellulose-based material with improved barrier properties provided the great potential as green and sustainable packaging materials.

## Figures and Tables

**Figure 1 polymers-11-01929-f001:**
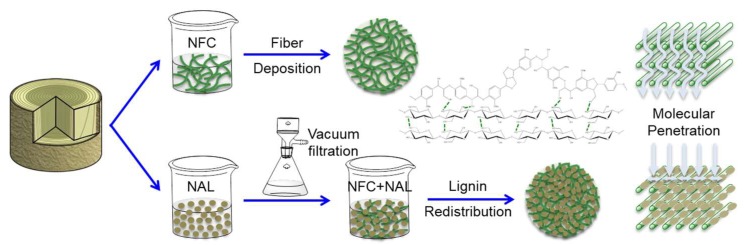
Lignin redistribution for enhancing barrier properties of cellulose-based materials.

**Figure 2 polymers-11-01929-f002:**
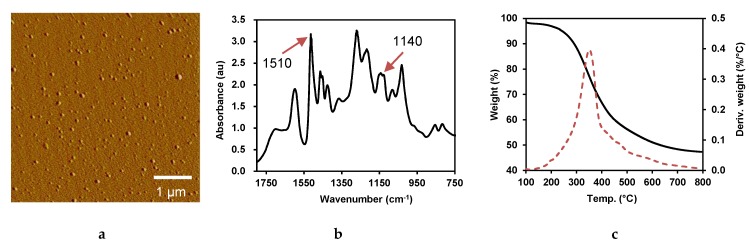
Characterization of nanolized alkali lignin ((**a**) atomic force microscopy (AFM), (**b**) Fourier transform infrared spectroscopy (FTIR), (**c**) thermogravimetric analysis (TGA)).

**Figure 3 polymers-11-01929-f003:**
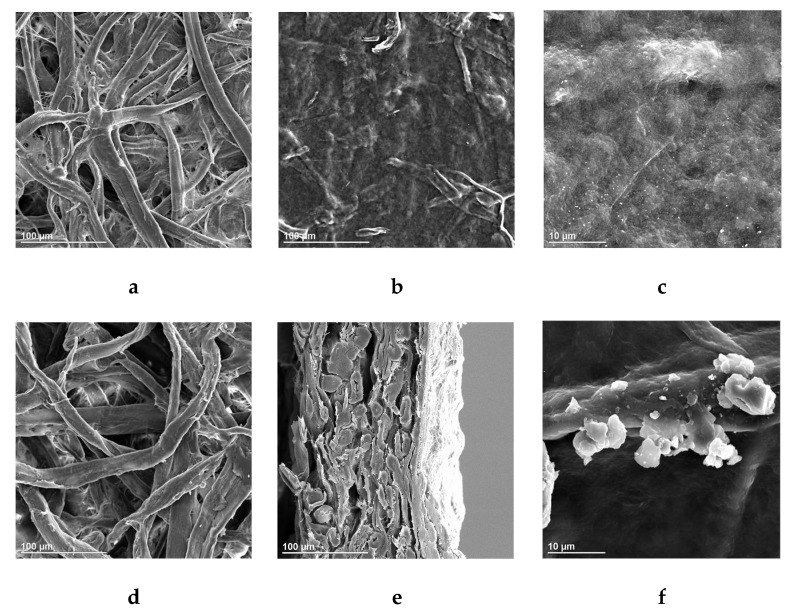
Scanning electron microscopy (SEM) images of deposited paper surface ((**a**) P4 (Control), scale: 100 μm, (**b**) P4 (NFC+NAL), scale: 100 μm, (**c**) P4 (NFC+NAL), scale: 10 μm, (**d**) P8 (Control), scale: 100 μm, (**e**) P8 (NFC+NAL), scale: 100 μm, (**f**) P8 (NFC+NAL), scale: 10 μm).

**Figure 4 polymers-11-01929-f004:**
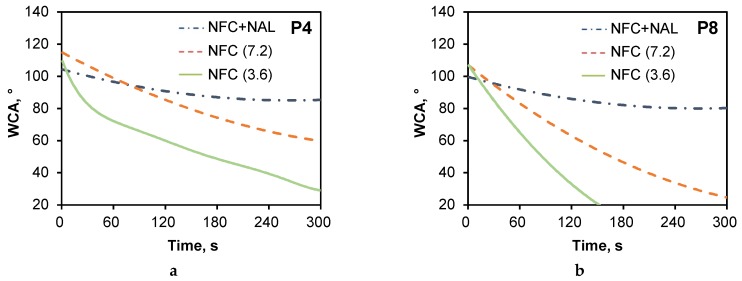
Time dependent water resistance of Nanolized alkali lignin (NAL) redistributed P4 (**a**) and P8 paper (**b**).

**Figure 5 polymers-11-01929-f005:**
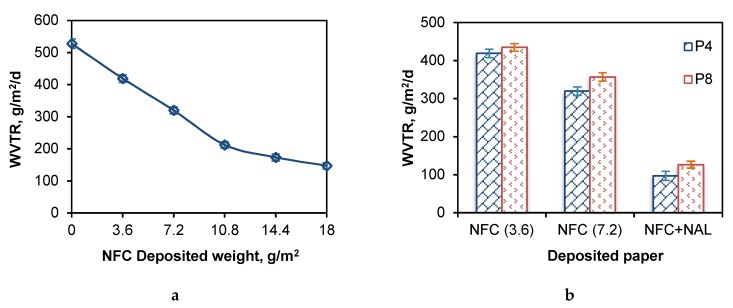
Water vapor transmission rate (WVTR) of nanofibrillated cellulose (NFC) deposited P4 paper (**a**) and NAL redistributed P4/P8 paper (**b**).

**Figure 6 polymers-11-01929-f006:**
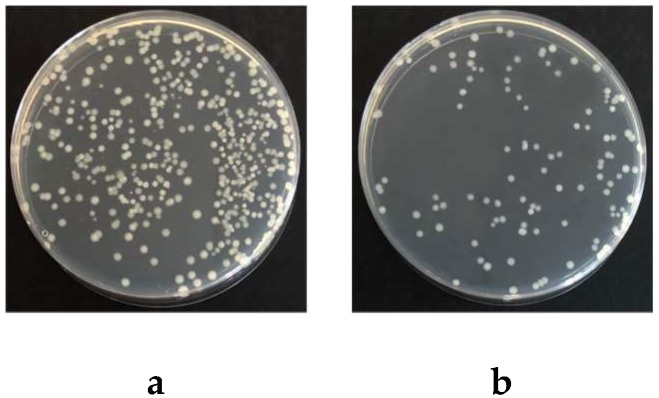
Pictures of the antibacterial activities of deposit paper against *E. coli* ((**a**) P4 (Control), (**b**) P4 (NFC + NAL)).

**Table 1 polymers-11-01929-t001:** Grease stained areas (%) of deposited paper.

Samples	Duration				
	4 h	8 h	24 h	48 h	72 h
P4 (Control)	16.6 ± 0.6	30.9 ± 0.9	62.4 ± 0.8	77.0 ± 0.9	90.5 ± 0.8
P4 (NFC-3.6 g/m^2^)	0.2 ± 0.2	0.5 ± 0.4	6.6 ± 0.6	18.1 ± 0.8	37.1 ± 1.6
P4 (NFC + NAL)	ND	ND	ND	ND	ND
P8 (Control)	20.6 ± 0.8	38.9 ± 0.7	72.9 ± 0.9	92.0 ± 0.8	99.3 ± 0.3
P8 (NFC-3.6 g/m^2^)	0.3 ± 0.1	0.8 ± 0.6	6.9 ± 0.7	20.4 ± 0.9	40.8 ± 1.2
P8 (NFC + NAL)	ND	ND	ND	ND	ND

Note: ND, Not detectable.

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
