# Peer review of "Lignin Redistribution for Enhancing Barrier Properties of Cellulose-Based Materials"

_polymers, 2019, doi:10.3390/polym11121929_

Round 1

Reviewer 1 Report

Lignin redistribution for enhancing barrier properties of cellulose-based materials

Comments

What was the loading capacity or ratio of NFC and NAL used? It is not clear. Data on polydispersity should be provided. Zeta potential analyzer was used to determine particle size?? Line 146. Statistical significance of results should be included in all the provided data. The value of n is missing in various studies like water contact angle measurement. What was the optimized deposition weight of NFC-NAL? The effect of the deposition weight of NFC-NAL on grease resistance is not incorporated. Standard error and statistical analysis are required for antibacterial studies. What is n? Using a single bacterial strain of E.coli for claiming the antibacterial activity is insufficient. Other common micro-organisms should also be considered. Providing safety profile of lignin used is essential. Leaching of lignin nanoparticles into food material can be a cause of concern. Thus leaching studies are needed. These papers should be cited https://doi.org/10.3390/nano9020243, https://doi.org/10.3390/biom9080363.

Major addition of data is required to support the claim made by the authors.

Author Response

Best,

Huining

Reviewer 2 Report

This paper describes the fabrication of lignin nanoparticles and nanocellulose based food packaging films with high performance barrier properties. Overall, the study is well designed and the experimental results presented in the context are convincing and support the conclusion. Also the content is suitable for publishing in Polymers. Here are a few suggestions.

In the introduction, author need to more explain the potential and advantages of nanolignin and nanocellulose with recently published papers, for example, New J. Chem 2018, 42, 3415, J. Mater. Chem. C, 2018, 6, 6423, ACS Sustainable Chemistry & Engineering 2019, 7, 15640, Nanomaterials 2019, 9, 612.                  

Authors should revise the Figure 1 to show the fabrication process (vacuum filtration) with nanolignin/nanocellulose solution in detail.

I recommend to add the cross-sectional SEM images to Figure 3. It might be helpful for demonstrating the structures of composite papers.

Author Response

Best,

Huining

Round 2

Reviewer 1 Report

The authors have addressed the reviewer's comments.